# New High-Rate Timestamp Management with Real-Time Configurable Virtual Delay and Dead Time for FPGA-Based Time-to-Digital Converters

Fabio Garzetti [†], Gabriele Bonanno [†], Nicola Lusardi *,[†] [ID], Enrico Ronconi, Andrea Costa and Angelo Geraci

Dipartimento di Elettronica, Informazione e Bioingegneria (DEIB), Politecnico di Milano,
Via Golgi 40, 20133 Milano, Italy; fabio.garzetti@polimi.it (F.G.); gabriele.bonanno@polimi.it (G.B.);
enrico.ronconi@polimi.it (E.R.); angelo.geraci@polimi.it (A.G.)
* Correspondence: nicola.lusardi@polimi.it
[†] These authors contributed equally to this work.

**Abstract:** Modern applications require the ability to measure time events with high resolution, a full-scale range, and multiple input channels. Time-to-Digital Converters (TDCs) are a popular option to convert time intervals into timestamps. To reduce the time-to-market and Non-Recurring Engineering (NRE) costs, a Field-Programmable Gate Array (FPGA) implementation has been chosen. The high number of requested bits and channels, however, gives rise to routing congestion issues when routed in a parallel manner. In this paper, we will propose and analyze a novel solution, the Belt-Bus (BB), which involves a parallel-to-serial conversion of the timestamp stream coming from the TDC while maintaining chronological order and a sufficient high rate, and flagging the presence of timestamp overflow. Moreover, two new useful features are added. The first is a "Virtual Delay" to compensate for offsets due to cable length and FPGA routing path mismatch. The second is a "Virtual Dead-Time" to filter out unforeseen events. Finally, the BB was tested on a Xilinx 28 nm 7-Series Kintex-7 325T FPGA, achieving an overall data rate of 199.9 Msps with very limited resource usage (i.e., lower than a total of 4.5%), consuming only 480 mW in a 16-channel implementation.

**Keywords:** Time-to-Digital Converter (TDC); data serialization; timestamp; Belt-Bus (BS); Field-Programmable Gate Array (FPGA)

## 1. Introduction

The Time-to-Digital Converter (TDC), which assigns a timestamp to an event, is a device used in various commercial and industrial settings, ranging from basic experimental setups to complex research and development projects [1–4]. Prominent examples of its major uses include Time-of-Flight Positron Emission Tomography (TOF-PET) [5] in the biomedical field and Laser Rangefinder [6] techniques for 3D imaging in industry and the automotive field. In the context of time-resolved spectroscopic experiments, TDCs are extensively utilized in academic settings, particularly in techniques like Time-Correlated Single Photon Counting (TCSPC) and the pump-and-probe experiment performed using a Free Electron Laser (FEL) or a synchrotron light [7].

The majority of modern 3D industrial image sensors employ a TDC system to measure the time it takes for a laser pulse to be detected after emission. These sensors, known as Light Detection and Ranging (LIDAR) or Time-of-Flight (TOF) sensors [8,9], find wide applications in areas such as aerial inspection [10] and autonomous driving [6]. Specifically, LIDARs [11] require a TDC with a large number of channels to benefit from a high frame rate, wide field of view, and excellent reliability [12].

Thanks to their cost-effective Non-Recurring Engineering (NRE) expenses, their excellent performance achieved, and their reprogrammable nature, FPGA-based TDC systems

stand out as an optimal solution for fast prototyping, both in the realm of research and in industrial Research and Development (R&D) [13].

All of the modern applications mentioned above, both academic and industrial, require TDCs with a high Full-Scale Range (FSR) and resolution (LSB); thus, with a high number of bits on a significant number of parallel channels and the ability to operate at high rates (e.g., tens of megahertz per channel) [14]. In addition, these timestamps must be processed immediately in real time by various modules working in parallel typically hosted in programmable logic devices like Systems-on-Chip (SoCs) and Field-Programmable Gate Arrays (FPGAs) posing a routing challenge between the TDC and processing modules, both in terms of the congestion of the routing itself (i.e., a high number of required wires) and the potential generation of Cross-Talk (XT) events .

Hence, there is a drive to conceive and develop a parallel-to-serial timestamp data transmission architecture to facilitate the routing of tens of bits (i.e., high FSR and resolution) in multi-channel systems. This is complicated by the need for the serialization process to maintain the chronological order of the timestamps generated by the TDCs while simultaneously managing overflow phenomena, which can significantly impact processing efficiency. As a solution, a novel high-efficiency parallel-to-serial timestamp data transmission architecture protocol based on the AXI4-Stream protocol [15] (also known as AXIS), named Belt-Bus (BB), has been fully developed and validated as an IP-Core in TDC architectures implemented in FPGAs. It is worth noting that this architecture is equally suitable for implementation in ASICs.

This paper is structured as follows: After a description of TDCs and an overview of multi-channel system interconnection, Section 3 describes the proposed protocol and structure, while Section 4 addresses the main issues and their respective resolutions. The final structure is outlined in Section 5. Characterization in terms of area occupancy, power dissipation, and performance along with measurements conducted on a 16-channel TDC implemented in a Xilinx 28 nm 7-Series Kintex 325-T FPGA, is presented in Section 6.

## 2. Time-to-Digital Converter

In Section 2.1, the main Figures-of-Merit (FoMs) of the TDC will be summarized; additionally, in Section 2.2, the issue of connections for multi-channel TDC systems will be illustrated, with a related overview of the state of the art.

### 2.1. Backgrounds

In the scientific literature and in the industrial field, there are various architectures of TDCs implementable in both ASICs [16] and FPGAs [13]. Regardless of the type of structure, a TDC assigns a timestamp, referring to the clock with which the TDC is powered, to the occurrence of a low–high and/or high–low transition on the inputs. Being a digital device, in addition to the temporal reference for the timestamp, the clock of the TDC serves to manage the internal logic. Regardless of the architecture and their implementation in programmable logic (i.e., FPGA, SoC) or ASIC, TDCs are characterized by the following FoMs:

- Resolution or LSB: the smallest time interval that can be accurately measured.
- Precision or Jitter: variation in the output timing accuracy of the TDC.
- Linearity: the degree to which the digital output is proportional to the input time interval, expressed as Differential and Integral Non-Linearity (DNL and INL).
- Full-Scale Range (FSR): the maximum time interval measurable without encountering overflow issues.
- Frequency of Overflow ($f_{ovfl}$).
- Number of bits ($N_{bit}$).
- Number of channels operating in parallel ($N_{CH}$).
- Dead Time (DT): the time that elapsed between two successive measurements on the same channel.

- Maximum Channel Rate (R): the maximum rate of measurements that a single channel can perform.
- Maximum Output Data Rate (ODR): the maximum rate of output processed timestamps.
- Area Occupation: physical size (for ASIC) or number of resources (for ASIC and FPGA) occupied.
- Power Consumption: the amount of power consumed.

From these FoMs, we can easily calculate some relationships that exist among them, such as the connection between FSR, LSB, and $N_{bit}$ (1), between FSR and $f_{ovfl}$ (2), between FSR, $N_{bit}$, and LSB (3), and the obvious inequalities that link DT with R (4), and ODR with R and Nch (5).

$$N_{bit} = \log_2(FSR/LSB) \tag{1}$$

$$f_{ovfl} = 1/FSR \tag{2}$$

$$FSR = 2^{N_{bit}} \cdot LSB \tag{3}$$

$$R \leq 1/DT \tag{4}$$

$$ODR \leq N_{CH} \cdot R \tag{5}$$

*2.2. Multi-Channel Connection Issues and State of the Art*

Considering a multi-channel TDC, regardless of its architecture and implementation (e.g., FPGA/SoC vs ASIC), different solutions for timestamp read-out can be employed: serial or parallel. The difference between the two approaches lies in the fact that in the parallel read-out, each channel has a dedicated output line for the timestamp, whereas in a serial solution, there exists an arbitration mechanism for serialization. The adopted approach is relatively insignificant if the number of channels is low but becomes crucial for systems with eight or more channels.

If we consider a parallel read-out approach with a high number of implemented channels, we will have a total of $N_{CH} \times N_{bit}$ lines to route within our device. This creates substantial internal congestion that severely limits place and route operations. Furthermore, this solution is inconvenient if the information needs to be transferred externally, as it would require a package with a high number of pins. Moreover, managing a large number of lines further increases the likelihood of generating XT events that interfere with sensitive parts of the circuitry. For this reason, an output serialization mechanism is incorporated into TDCs with a high number of channels. The effectiveness of such a circuit will significantly impact various FoMs of the TDC, such as the ODR, DT, R, area occupancy, and power consumption.

Indeed, a non-optimized and simpler serialization and sorting mechanism such as a round-robin (e.g., Timepix3 ASIC-TDC) algorithm performs well in terms of the ODR but not in terms of area and power consumption [17,18]. In this context, a system with $N_{CH}$ channels requires, approximately, a multiplexer with $N_{CH}$ inputs (i.e., $N_{CH}$-to-1 MUX), whose area occupation and power dissipation exponentially increase with $N_{CH}$ [3]. On the other hand, there are serialization systems that, to keep power consumption and area low, rely on memories (e.g., PicoTDC) that record all timestamps for a certain acquisition time and then serially output them. Some of them, however, have high DTs and low rates (e.g., PETsys) [19,20], while others output timestamps without any order and sorting [21,22], requiring an additional processing stage downstream of the TDC if real-time processing is required by various modules working in parallel, such as histograms, counters, and coincidence detectors [4].

The proposed BB solution consists of an innovative serialization structure based on timestamp sorting through comparison, similar to what happens in round-robin. However, the distinctive feature is the distribution of the comparison process on 2-to-1 MUX distributed within the $N_{CH}$ nodes. This allows for high efficiency in terms of area occupancy and power consumption that scale linearly with $N_{CH}$. Furthermore, the presence of memories and pipeline structures enables high data acquisition rates (ODR) to be achieved without compromising DT and R.

## 3. The Belt-Bus

In Section 3.1, the BB protocol is explained; the operating principles are described in Section 3.2, while a detailed logical description of the functioning is presented in Section 3.3, analyzing the submodules. The area occupancy and power dissipation of each submodule are presented in Section 3.4, with the Xilinx 28 nm 7-Series Kintex 325-T FPGA used as a case study.

### 3.1. Protocol

The BB is a synchronous bus based on AXI4-Stream and utilizes only the TVALID, TREADY, and TDATA signals. As a convention, a logical one for both TVALID and TREADY signifies a valid TDATA. The TDATA signal, as illustrated in Figure 1, comprises three portions: the Timestamp (TS) field with an obvious dimension of $N_{bit}$, a 2-bit wide Function Identifier (FID) field, and the Number of Channel (NUM_CH) field. The latter field has a non-defined a priori dimension to appropriately accommodate the number of channels involved in the measurement (i.e., $\log_2(N_{CH})$), representing the channels' numerical value.

The architecture of the BB was also designed to address the main issues related to the operation of the TDC without modifying the number of bits of the timestamps, specifically addressing overflow concerns. This was achieved through the utilization of the FID and TS, providing information to downstream modules about particular characteristics deemed useful in subsequent processing.

In the implementation presented here, the FIDs are coded as follows:

- FID = 00: overflow event (in TS the overflow value is sent);
- FID = 01: timestamp coming from a rising edge event;
- FID = 10: unused;
- FID = 11: timestamp coming from a falling edge event.

Each time an overflow occurs, a new frame with FID = 00 is injected into the BB. Now the FSR can be increased by a factor $2^{N_{bit}}$ from $2^{N_{bit}} \cdot LSB$ up to $2^{N_{bit}} \times (2^{N_{bit}} \cdot LSB)$. However, this improvement comes at a cost to the Output Data Rate (ODR), as an overflow event must be sent once every $2^{N_{bit}} \cdot LSB$ instead of the current timestamp. With the overflow frequency denoted as $f_{ovfl} = 1/(2^{N_{bit}} \cdot LSB)$ and $f_{CLK,BB}$ as the clock frequency of the BB, the rate is given by:

$$ODR = f_{CLK,BB} - N_{CH} \cdot f_{ovfl} = f_{CLK,BB} - N_{ch} \frac{1}{2^{N_{bit}} \cdot LSB} \tag{6}$$

Equation (6) shows that there is a trade-off regarding this aspect. By decreasing $N_{bit}$, the ODR also decreases. It is important to notice that this trade-off is heavier as the number of channels increases. However, this depends on the used FPGA's size. With larger and more complex FPGAs, routing issues can be minimized, enabling a slight increase in $N_{bit}$ and, in most cases, $f_{CLK,BB}$, thus enhancing the available data rate.

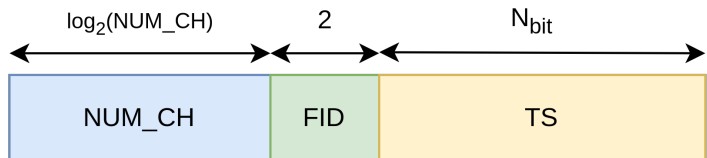

**Figure 1.** Fields of BB in TDATA signal.

### 3.2. Principle of Operation

With each node representing a single channel, the BB is composed of a cascade of nodes (light blue in Figure 2) that serialize, in a pipeline way, the timestamp coming from TDC channels (yellow in Figure 2) in a chronological sequence. Every node has two inputs in BB protocol: the output of the preceding node, the "Top" port in Figure 2, and the current channel, the "Left" port in Figure 2. The only restriction on the number of channels that

may be added with this chain arrangement is the amount of hardware that can be used by implementing the nodes and the constraint on the average channel rate (i.e., the ratio between the ODR and the total number of channels).

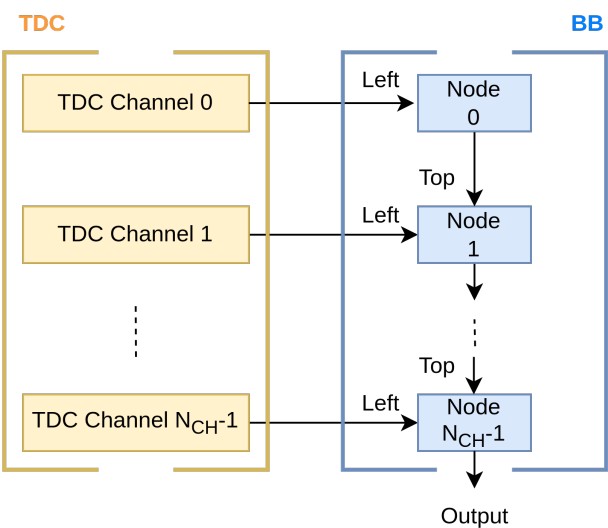

**Figure 2.** Structure of the Belt-Bus.

When a timestamp arrives at a node from the channel entrance (also known as "Left" port), it propagates through subsequent nodes to the terminal one. While timestamps can be arranged chronologically within a single channel, as they move through the chain of nodes, that arrangement may be lost. To prevent this, based on which of the two timestamps is temporally earlier, each node decides whether to prioritize the input from the channel (also known as "Left") or that from the previous node (also known as "Top"). It might not be feasible to compare the input timestamp in the present node, though, because of potential delays in measurements on channels connected to earlier nodes and, consequently, the absence of the comparative timestamp. In this instance, the current timestamp could be propagated without adhering to the chronological order. Of course, the first node allows only the injection of the "Left" signal, so it possesses the "Top" signal with TVALID hardcoded to "0".

In order to address this problem, there are four phases involved in obtaining the timestamp from the input channel through the node. Considering the contextualization of their dynamics in the architecture described in the next paragraph, from the perspective of their respective functions, these phases are, as follows, the:

1. Retain Phase: the timestamp from the TDC channel (also known as "Left" port) is blocked for a proper time at the input of the node in order to compensate for the pipeline introduced by the registers and FIFO present on the previous nodes.
2. Hold-on Phase: the timestamp from the TDC channel (also known as "Left" port) waits for a timestamp from the previous node for comparison for a finite time. If this occurs, the older timestamp is propagated at the node output.
3. Inject Phase: if the timestamp from the previous node (also known as "Top" port) is not valid or older, the timestamp from the channel (also known as "Left" port) is propagated at the node output.
4. Discard Phase: the timestamp from the TDC channel (also known as "Left" port) is simply discarded (not propagated in BB) because there is no propagation permission in the node chain within a finite time (for instance, if the chain were full).

For example, a graphical view of the operation of these phases on three timestamps is shown in Figure 3.

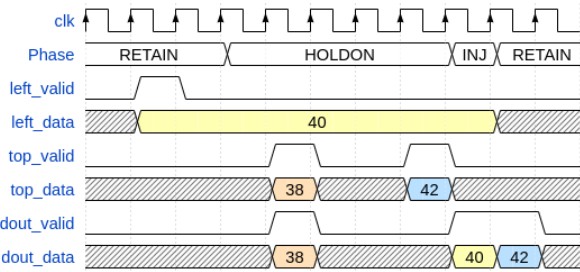

**Figure 3.** Timing diagram showing the phases through which the timestamps 38, 40, and 42 enter the node chain.

### 3.3. Architecture

The designed architecture of the BB was implemented on a Xilinx 28 nm 7-Series FPGA as an IP-Core and constituted by a cascade of stages called Node Inserters (Figure 4).

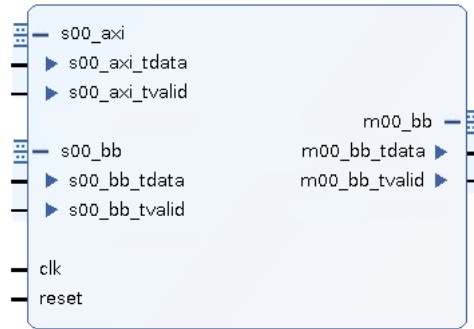

**Figure 4.** Node Inserter IP-Core; m00_bb is the output port and s00_axi and s00_bb are the "Left" and "Top" input ports, respectively.

With reference to Figure 5, three components go into making up each Node Inserter; i.e.,

- The Delay Synchronizer implements the Retain Phase shown in Section 3.2.
- The Inserter is driven by logic that, using the information from the Delay Synchronizer, generates the selection signal for a multiplexer between the timestamp from the current channel (also known as "Left" port) and the one coming from the previous node (also known as "Top" port). Thus, it implements the Hold-on, Inject, and Discard Phases shown in Section 3.2.
- The Super Sampler, which is a register that propagates the selected input to the output, ensuring the ready–valid handshake proper to the AXI4-Stream protocol without losing a clock cycle.

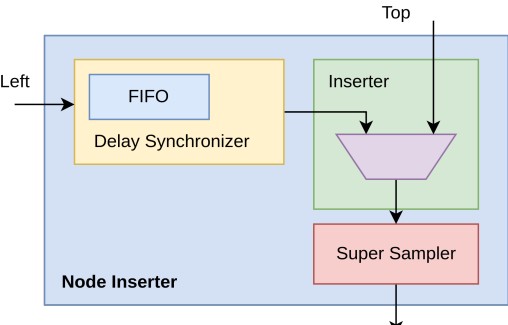

**Figure 5.** Top level block schematic of Node Inserter structure with submodules.

The Delay Synchronizer makes the current timestamp from the TDC channel (also known as "Left" port) comparable to the one coming from the preceding node (also known as "Top" port). First of all, the incoming timestamp from the TDC channel (also known

as "Left") enters into a synchronous First-In First-Out (FIFO) clocked at $f_{CLK,BB}$ hosted in the Synchronizer. The validity of the output data is deasserted, preventing its propagation, until the propagation time through the FIFO has elapsed (i.e., Retain Phase). The TREADY signal at the "Top" interface allows the information to be stored in the FIFOs and registers of the previous nodes, thus avoiding the presence of an additional FIFO.

Due to the potential for skew and jitter phenomena when signals spread over a wide region, from a timing perspective, a highly intricate and sophisticated data management system is required inside the Delay Synchronizer to ensure that timestamp values can be compared. Thus, the Inserter module behaves as a multiplexer driven by logic that compares the current timestamp (also known as "Left" port) with the one coming from the preceding node (i.e. "Top" port) based on information returned by the Delay Synchronizer and moves ahead with overflows with the highest possible priority, followed by timestamps from the oldest to the newest. If the bandwidth saturates, newer timestamps are discarded. In detail, in the Inserter, if an overflow condition is communicated from the timestamp at the output of the Delay Synchronizer (also known as "Left" port) or from an older timestamp present from the previous node (also known as "Top" port), the incoming timestamp (i.e., the output of the Delay Synchronizer) is propagated forward. The assessment of this condition continues until the timeout, equal to an interval comprising the clock jitter, skews, and the delays of the pipeline stages that constitute the implementation (i.e., Hold-on Phase). At the timeout of the Hold-on Phase, if the node's bus is ready to receive, the timestamp is propagated (Inject Phase); otherwise, it is discarded, allowing for a more recent timestamp to be placed at the FIFO output (Discard Phase).

### 3.4. Area Occupancy and Power Dissipation

The area occupation of the Node Inserter and its related submodules is a function of the number of bits in the TS fields (i.e., $N_{bit}$). Table 1 presents the area occupancy in terms of Carry Logic (CARRY), Look-Up Tables (LUT), Flip-Flops (FF), and Look-Up Table RAM (LUTRAM) occupied. No resources in terms of Digital Signal Processor (DSP) modules and Block RAM (BRAM) are utilized. Additionally, the same table provides information on power dissipation (only dynamic considering that the power dissipated by the module is primarily of a dynamic nature), considering a maximum clock frequency of 130 MHz.

**Table 1.** Area occupancy and power dissipation of Node Inserter and its related submodules clocked at 130 MHz, as presented in Section 3.

| $N_{bit}$ | Module/Submodules | Power [mW] | CARRY | LUT | FF | LUTRAM |
|---|---|---|---|---|---|---|
| | Node Inserter | 7 | 8 | 137 | 148 | 20 |
| 16 | Delay Synchronizator | 5 | | 126 | 64 | 20 |
| | Inserter | 1 | 8 | 2 | 2 | |
| | Super Sampler | 1 | | 9 | 82 | |
| | Node Inserter | 7 | 12 | 149 | 180 | 24 |
| 24 | Delay Synchronizator | 5 | | 139 | 114 | 24 |
| | Inserter | 1 | 12 | 2 | 2 | |
| | Super Sampler | 1 | | 8 | 64 | |
| | Node Inserter | 9 | 16 | 169 | 212 | 28 |
| 32 | Delay Synchronizator | 7 | | 158 | 128 | 28 |
| | Inserter | <1% | 20 | 2 | 2 | |
| | Super Sampler | 2 | | 9 | 82 | |

**Table 1.** *Cont.*

| $N_{bit}$ | Module/Submodules | Power [mW] | CARRY | LUT | FF | LUTRAM |
|---|---|---|---|---|---|---|
| 40 | Node Inserter | 10 | 20 | 234 | 244 | 36 |
| | Delay Synchronizator | 8 | | 223 | 144 | 36 |
| | Inserter | <1% | 20 | 2 | 2 | |
| | Super Sampler | 2 | | 9 | 98 | |
| 48 | Node Inserter | 11 | 24 | 264 | 276 | 40 |
| | Delay Synchronizator | 9 | | 253 | 150 | 40 |
| | Inserter | <1% | 24 | 2 | 2 | |
| | Super Sampler | 2 | | 9 | 124 | |
| 56 | Node Inserter | 13 | 28 | 292 | 308 | 44 |
| | Delay Synchronizator | 9 | | 281 | 176 | 44 |
| | Inserter | 1 | 28 | 2 | 2 | |
| | Super Sampler | 2 | | 9 | 130 | |
| 64 | Node Inserter | 14 | 32 | 324 | 340 | 48 |
| | Delay Synchronizator | 10 | | 313 | 189 | 48 |
| | Inserter | 1 | 32 | 2 | 2 | |
| | Super Sampler | 3 | | 9 | 149 | |

## 4. Main Issues and Solutions

As presented in Section 3, the BB also has two limitations.

The principal issue is that the present structure does not fully account for the uncertainty of the timestamp arrival time. In fact, two timestamps produced from distinct channels during the same TDC clock cycle can arrive at the Node Inserter at different times. This is particularly true at high channel rates. The primary cause of this is the needs of asynchronous FIFOs to accommodate the Clock Domain Crossing (CDC) between the clock of the TDC ($f_{CLK,TDC}$) and the clock of the BB ($f_{CLK,BB}$). This establishes the likelihood of unordered timestamps in specific scenarios. The solution to these issues is addressed in Section 4.1.

Another issue is that, considering the Xilinx 28 nm 7-Series FPGAs as a technological node, due to the architecture presented in Section 3, the maximum available BB clock frequency is not very high, about 130 MHz (i.e., $f_{CLK,BB} < 130$ MHz), which corresponds to only 16.25% of the maximum clock frequency that these technological nodes support (i.e., 800 MHz). The cause of these issues is analyzed and discussed in Section 4.2. Thanks to these two modifications, a frequency of 200 MHz (25% of the maximum available) can be achieved. The area occupancy and power dissipation of each submodule are presented in Section 4.3.

### 4.1. Unsorted Timestamps Issue

The first issue that has been addressed is related to the presence of different clock domains, which require asynchronous FIFOs between the TDC and BB as a CDC (Figure 6).

Under this condition, there are two further causes of timestamp unsorting. The first one, deterministic, is due to the different clock frequencies in case one channel has a high timestamp rate and another has a lower one; a timestamp entering in the first asynchronous FIFO can exit from it in a different time instant and so is injected late in the BB with respect to the other one in a less crowded channel since the data already stored in the asynchronous FIFO must exit first.

The second one is non-deterministic and is due to unpredictable CDC propagation delay. To better understand this, let us focus briefly on how a CDC works in the following subparagraphs.

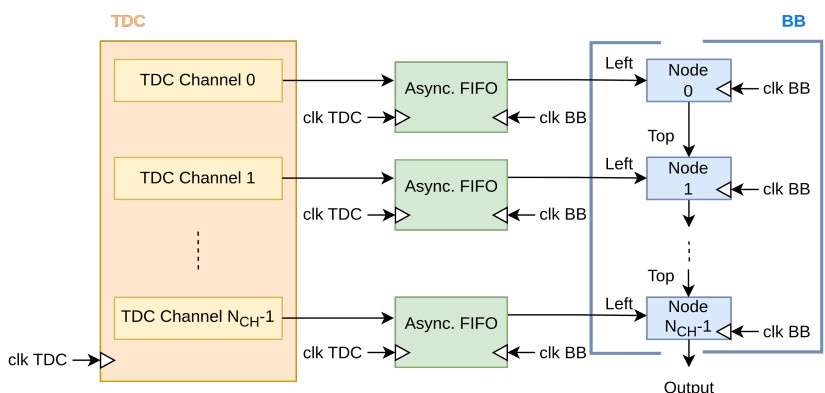

**Figure 6.** Connection between the TDC (orange), clocked at clk TDC, and the BB (blue), clocked at clk BB, is established using an asynchronous FIFO (Async. FIFO, green) employed as CDC.

### 4.1.1. CDC Uncertainty

In a basic CDC circuit, the simplest 1-bit two-stage architecture can be considered (Figure 7). In the first stage (flip-flop A), data are captured by a register in the source clock domain on the rising edge of the source clock (i.e., clk A). In the second stage (flip-flop B), the captured data are then transferred to a register in the destination clock domain on the rising edge of the destination clock (i.e, clk B). The clock uncertainty in this circuit arises because the rising edges of the source and destination clocks may not be perfectly aligned in time due to factors such as clock skew, jitter, or delay. As a result, data may be captured by the source register at a slightly different time than they are transferred to the destination register.

As consequence of that, the sampling register could enter into metastability [23]; on average, each Mean Time Between Failures (MTBF) given by the relation $MTBF = \frac{f_r}{t_0 \cdot f_{CLK,A} \cdot f_{CLK,B}}$, where $f_r$ is a parameter that depends on the flip-flop used, $t_0$ is a constant related to the width of the time window or aperture wherein a data edge triggers a metastable event, $f_{CLK,A}$ is the source clock domain frequency, and $f_{CKL,B}$ is the destination clock domain frequency. To quickly exit a possible metastability transient, the well-established cascade of registers must be added. Now, if, for example, two registers are put in cascade, there is not only one clock uncertainty due to sampling but another one, with lower probability, needed from the first register to recover from metastability. This uncertainty increases as the number of cascaded registers grows. Even in an asynchronous FIFO, if the two clocks are not derived from the same source (e.g., a divided clock), similar mechanisms are used internally, giving rise to a temporal uncertainty at the FIFO output.

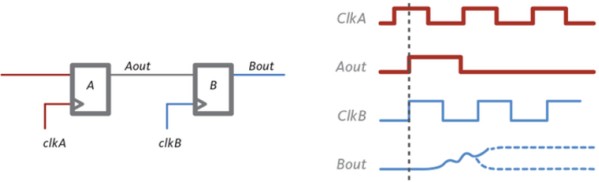

**Figure 7.** Basic Clock Domain Crossing structure and waveforms.

### 4.1.2. Issue Evidence

By examining simulated waveforms focusing on two channels for simplicity, two situations emerge, resulting in unsorted timestamps. Figure 8 shows these two situations. In the first case (on the top side of Figure 8), timestamps "12" and "13" are sent to the "Left" port of the relative Node Inserter at the same instant, but due to the CDC issue, timestamp "13" is read before timestamp "12", resulting in an unordered timestamp error on BB. In the second case (on the bottom side of Figure 8), timestamp "33" stays, due to CDC uncertainty,

in the asynchronous FIFO for more time compared to timestamp "34", causing unordered issues at the output.

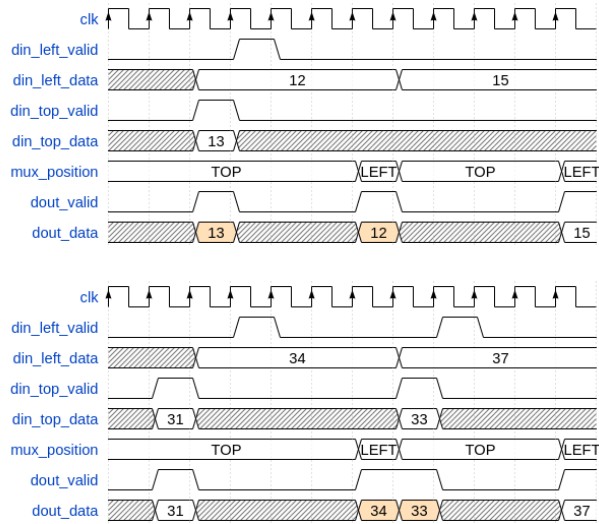

**Figure 8.** Waveforms with CDCs modeled.

### 4.1.3. Issue Solving

In order to mitigate this failure, two modifications have been introduced in the Node Inserter.

Since the timestamp entering the Node Inserter cannot be injected until the Hold-on Phase is active unless a newer timestamp reaches the "Top" port, the Retain Phase on the Delay Synchronizer is increased by a value larger than the time uncertainty introduced by the asynchronous FIFO used as a CDC, allowing the data to be properly compared by the Inserter. This way, the issue arrived on the top in Figure 8 is solved.

The second modification to the Inserter is mandatory to solve the issues present on the bottom of Figure 8. Instead of sending data to the Super Sampler when the Inject phase begins, another check is performed by simply waiting some clock cycles after the timestamp coming from the previous node is propagated. In this way, when the bus is full, the data comparison is always performed, avoiding unsorted timestamps. The number of cycles to wait is proportional to the node number to compensate for pipelines and the ratio between $f_{CLK,TDC}$ and $f_{CLK,BB}$ (where $f_{CLK,TDC} > f_{CLK,BB}$). No wait cycles are requested if $f_{CLK,TDC} < f_{CLK,BB}$.

### 4.1.4. Order Checker

After the modifications introduced in Section 4.1.3, the chronological order issue becomes very rare and thus almost negligible: fewer than one in a billion samples (i.e., $1 \times 10^{-9}$). This residual error is due to the stochastic nature of the CDC (i.e., MTBF), especially when the number of channels is high, and events occur randomly and at a high rate; occasionally, unsorted timestamps may be present. A possible way to solve the problem could be to increase the asynchronous FIFO depth inside the Delay Synchronizer excessively, leading to area occupancy problems in the FPGA. To avoid this issue, given the very low probability of encountering unsorted data, these instances are simply discarded, resulting in a negligible loss.

To perform the chronological order check, another IP-Core has been developed, the Order Checker, which takes as input the data from the last Node Inserter and checks timestamp sorting, deasserting the validation if incoming data do not respect this condition.

As can be seen from Figure 9, the order checker has AXI4-Stream input (i.e., s00_bb in Figure 9) and AXI4-Stream output (i.e., M00_bb in Figure 9) for BB data, along with an AXI4 Memory-Mapped port to read out the number of unsorted timestamps, solely for debugging purposes (i.e., S00_axi in Figure 9). This module checks the incoming timestamp

and compares it with the already stored one: if the new one is more recent, the data are propagated and replace the already stored one; if it is older, the valid signal is deasserted, and the relative counter is incremented by one. If no data are present, i.e., the module has been initialized, the first timestamp is stored and propagated.

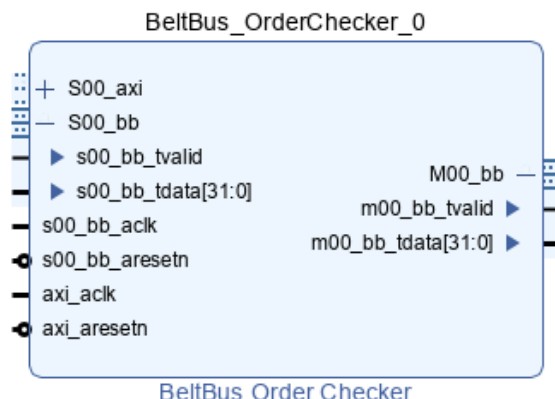

**Figure 9.** Order Checker IP-Core.

### 4.2. Limited Output Data Rate Issue

By implementing the Node Inserters in Vivado, considering the Xilinx 28 nm 7-Series FPGAs as a technological node, it can be clearly seen that the maximum clock frequency is limited by two different sources.

The main one comes from the way of performing the comparison (i.e., the symbols ">" in VHDL code) between the two timestamps entering the node (i.e., those coming to the "Left" and those coming from the "Top" ports), which requires, by the default encoding performed by Vivado, the computation of two subtractions and an unsigned comparison. This results in a very high requirement in terms of logic resources, mainly LUTs and CARRY, because the number of bits of the timestamps is high (e.g., 32 to 64). The intervention carried out to increase the maximum clock frequency was replacing the comparison operation (i.e., the symbols ">" in VHDL code) with a simpler one. Only a signed subtraction between the timestamps (i.e., "Left" minus "Top" in VHDL code) is performed, and then, a check is performed on the sign of the result. If the result is positive, the "Top" timestamp is older (i.e., "Left" is bigger than "Top" so more recent in time) and has the priority; otherwise, the "Top" data are propagated. Since performing the sign check is enough to observe the MSB of the result, the number of Carry Logic decreases by a factor of two, as they are only needed to perform one operation instead of three.

The second improvement can be introduced by replacing the Super Sampler with a more efficient pipelined structure, called AXIS Register Slice, that occupies the same hardware resources. The working principle is similar to having two-slot FIFOs. The data entering the module are stored in the output register if nothing is already stored in it. Thanks to these two modifications, a frequency of 200 MHz (20% of the maximum available) can be achieved.

### 4.3. Area Occupancy and Power Dissipation

The area occupation of the Node Inserter and its related submodules (with the modification proposed in this Section) is a function of the number of bits in the TS fields (i.e., $N_{bit}$). Table 2 presents the area occupancy in terms of CARRY, LUT, FF, and LUTRAM occupied. No resources in terms of DSP and BRAM are utilized. Additionally, the same table provides information on power dissipation, considering a maximum clock frequency of 200 MHz. Comparing Table 2 to Table 1, it is possible to observe a similar occupation and an increase by a factor of two in the CARRY occupied by the Inserter, along with the replacement of the Super Sampler with the AXIS Register Slice. Moreover, a higher usage of LUTs and FFs is

observed in the Delay Synchronizer to address the issues outlined in Section 4.1. The higher power dissipation is attributed to a higher clock frequency (200 MHz instead of 130 MHz).

**Table 2.** Area occupancy and power dissipation of Node Inserter and its related submodules presented in Section 4.

| $N_{bit}$ | Module/Submodules | Power [mW] | CARRY | LUT | FF | LUTRAM |
|---|---|---|---|---|---|---|
| **16** | Node Inserter | 10 | 4 | 128 | 192 | 20 |
| | Delay Synchronizator | 8 | | 127 | 108 | 20 |
| | Inserter | <1% | 4 | 2 | 2 | |
| | AXIS Register Slice | 2 | | 9 | 82 | |
| **24** | Node Inserter | 10 | 6 | 150 | 240 | 24 |
| | Delay Synchronizator | 8 | | 140 | 174 | 24 |
| | Inserter | 1 | 6 | 2 | 2 | |
| | AXIS Register Slice | 1 | | 8 | 64 | |
| **32** | Node Inserter | 13 | 8 | 170 | 288 | 28 |
| | Delay Synchronizator | 10 | | 159 | 204 | 28 |
| | Inserter | 1 | 8 | 2 | 2 | |
| | AXIS Register Slice | 2 | | 9 | 82 | |
| **40** | Node Inserter | 15 | 10 | 236 | 336 | 36 |
| | Delay Synchronizator | 12 | | 224 | 236 | 36 |
| | Inserter | <1% | 10 | 2 | 2 | |
| | AXIS Register Slice | 3 | | 9 | 98 | |
| **48** | Node Inserter | 17 | 12 | 265 | 384 | 40 |
| | Delay Synchronizator | 13 | | 254 | 258 | 40 |
| | Inserter | <1% | 12 | 2 | 2 | |
| | AXIS Register Slice | 4 | | 9 | 124 | |
| **56** | Node Inserter | 19 | 14 | 293 | 432 | 44 |
| | Delay Synchronizator | 14 | | 282 | 300 | 44 |
| | Inserter | 1 | 14 | 2 | 2 | |
| | AXIS Register Slice | 4 | | 9 | 130 | |
| **64** | Node Inserter | 21 | 16 | 325 | 480 | 48 |
| | Delay Synchronizator | 16 | | 314 | 329 | 48 |
| | Inserter | <1% | 16 | 2 | 2 | |
| | AXIS Register Slice | 5 | | 9 | 149 | |

## 5. Main New Features

The description of two new features, the Virtual Delay in Section 5.1, and Virtual Dead Time in Section 5.2, is the purpose of this section. These two improvements are performed with very careful attention to the timing analysis to ensure a maximum clock frequency of 200 MHz, as described in Section 4.2. Lastly, an overview of the complete structure of the BB is shown in Section 5.3. The area occupancy and power dissipation of each submodule are presented in Section 5.4

### 5.1. Virtual Delay

In many applications, only relative times must be measured by computing differences between different channels. For this reason, static offset compensation can be very useful, for example, to have the resulting histogram centered at zero. In order to perform this task, a Virtual Delay feature has been developed in the BB. In this way, the timestamps coming out from the Node Inserters are not only chronologically ordered but also translated in time. This allows compensating offsets due to both different cable lengths and FPGA routing path mismatches between TDC channels.

### 5.1.1. Architecture

A simple summing of the incoming timestamp delay is not sufficient to accomplish this feature. Indeed, if some synchronization mechanism is not present, the BB would "brake", leading to unsorted timestamps. As will be explained later, since synchronization requires memory, the Virtual Delay cannot reach very high values (i.e., up to $2^{20} \times \text{LSB}$), otherwise, the resource usage in the FPGA would be enormous. On the other hand, since the static offset due to FPGA routing is a few tens of picoseconds, considering an LSB of tens of femtoseconds, implementing a maximum delay in the order of hundreds of nanoseconds, which is quite feasible, would be enough. For example, by approximating the speed of signals at 30 cm/ns, a 1 µs delay would be sufficient to compensate for a 300 m cable length offset, which is a very high value. This is the main reason because the maximum delay value is less than or equal to the maximum timestamp value.

However, this is not the only thing that must be managed very carefully. When summing a value to a timestamp, the result can be larger than the maximum value of $2^{N_{bit}} - 1$. In this case, an overflow has to be generated, and careful attention must be paid to discard the next incoming one. With that said, another reason for choosing the maximum timestamp value is that the maximum overflow difference between the original and the delayed sample is one, which makes the process much simpler to implement.

Moreover, for the implementation of the Virtual Delay feature, a modular architecture has been used. A new module, the Virtual Delay Inserter, has been developed and instantiated in series before a modified version of the Delay Synchronizer called Virtual Delay Synchronizer. In detail:

- The Virtual Delay Inserter handles the summation between the delay and the timestamp. It is also responsible for overflow handling when overflows must be generated or discarded.
- The Virtual Delay Synchronizer handles the synchronization of the delayed timestamps.

Figure 10 shows the modular structure of the Node Inserter with the Virtual Delay functionality.

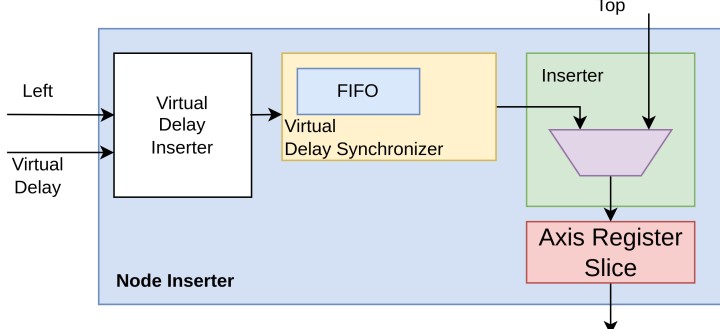

**Figure 10.** Modular structure of the Node Inserter with the Virtual Delay functionality.

### 5.1.2. Virtual Delay Inserter

This module consists of two pipeline stages.

The first stage is responsible for the timestamp computation. Since the summing of the delay can introduce an overflow, the second stage is needed to handle the overflow generation and the correct sampling and propagation of the timestamp, in order not to lose data. Since when an overflow is generated the next received one must be discarded, the data that are overwritten by the generated one are stored in a register. After this event, the timestamps are propagated through this register until there are no valid data to be sent. In cases where the rate is at maximum, this happens when an overflow is received from the TDC. Since the Virtual Delay can vary over time, another possible issue arises: if an overflow has been generated and the delay value decreases, the new timestamp can have a value that refers to the previous one. In order to solve this issue when an overflow is generated, the virtual delay, if it is lower than the previous one, is updated only after an

overflow from the TDC has been received. Finally, the generated overflow flag is needed by the Virtual Delay Synchronizer.

### 5.1.3. Virtual Delay Synchronizer

Compared to the previously introduced Delay Synchronizer, the Virtual Delay Synchronizer converts the Virtual Delay into a pulse of clocks at the BB clock ($\Delta VD$) to wait before starting the Retain Phase in order to synchronize the delayed timestamp injection into the BB.

### 5.2. Virtual Dead Time

The ability to insert a programmable dead time between measurements on the same channel is a really helpful feature. When a signal from a detector has a rising and/or a falling edge, a TDC timestamp is produced. Although filtering is typically requested, in fact, some input noise can still exist and cause unforeseen timestamps (red in Figure 11). These spurious timestamps could be discarded in post-processing by the elaboration modules; however, when the BB rate is high, such discarding might lead to saturations and result in the loss of samples. The Virtual Dead-Time functionality, which stops the incoming events for a programmable period of time (i.e., Virtual Dead Time, represented as keyword KILL in Figure 11) after one has been received, has been added to prevent this.

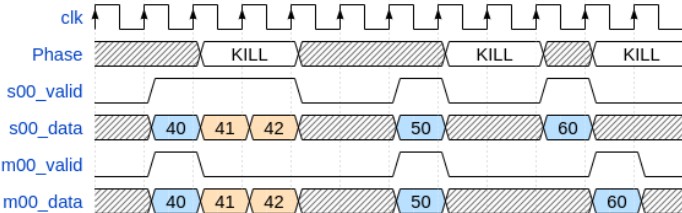

**Figure 11.** Waveform of the Virtual Dead-Time functionality.

A new IP-Core, named Time Killer (Figure 12), has been developed to enable this feature. The output valid is deasserted if the difference between the receiving timestamp and the input is smaller than the Virtual Dead-Time value. The IP-Core accepts timestamp form TDC (S00_AXIS input port Figure 12), a Virtual Dead-Time value, and provides a timestamp to the left port of the Node Inserter (M00_AXIS port in Figure 12).

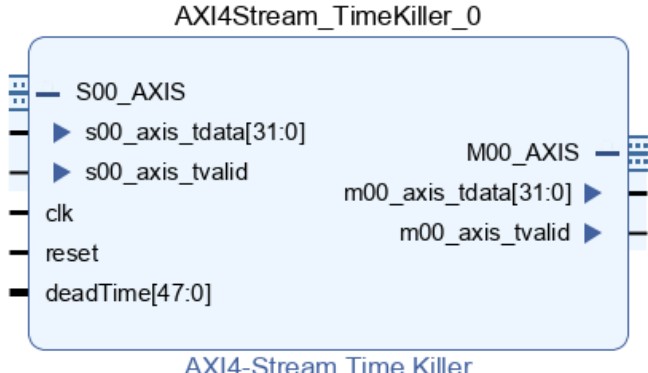

**Figure 12.** Time Killer IP-Core.

### 5.3. Final Belt-Bus Structure

In conclusion, the new BB structure's design, including the Virtual Delay and Virtual Dead Time functions, is shown in Figure 13.

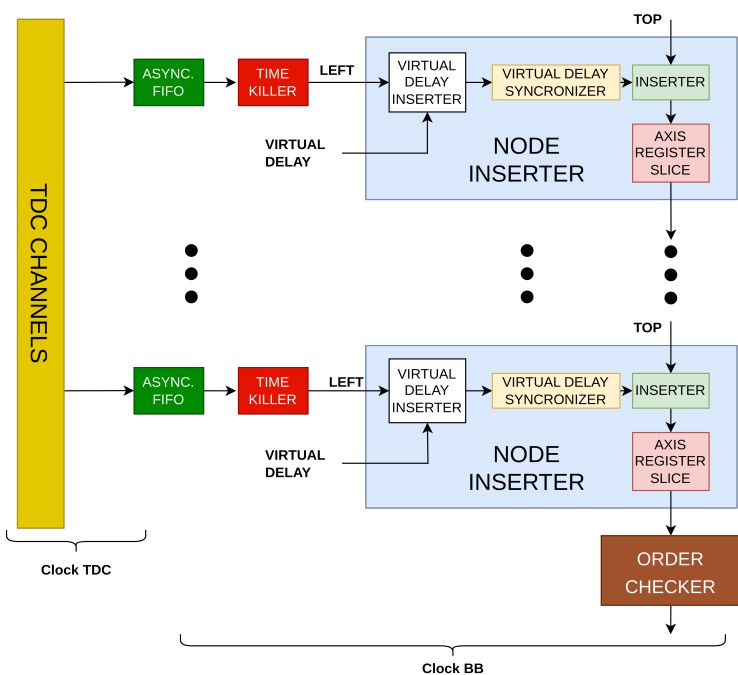

**Figure 13.** Structure of the improved BB.

*5.4. Area Occupancy and Power Dissipation*

The area occupation of the Node Inserter and its related submodules (with the modification proposed in this Section) is a function of the number of bits in the TS fields (i.e., $N_{bit}$). Table 3 presents the area occupancy in terms of CARRY, LUT, FF, and LUTRAM occupied. No resources in terms of DSP and BRAM are utilized. Additionally, the same table provides information on power dissipation, considering a maximum clock frequency of 200 MHz. Comparing Table 3 to Table 1, it is possible to observe a similar occupation and an increase by a factor of two in the CARRY occupied by the Inserter, along with the replacement of the Super Sampler with the AXIS Register Slice. Moreover, a higher usage of LUTs and FFs is observed in the Delay Synchronizer to address the issues outlined in Section 4.1. The higher power dissipation is attributed to a higher clock frequency (200 MHz instead of 130 MHz).

**Table 3.** Area occupancy and power dissipation of Node Inserter and its related submodules presented in Section 5.

| $N_{bit}$ | Module/Submodules | Power [mW] | CARRY | LUT | FF | LUTRAM |
|---|---|---|---|---|---|---|
|  | Node Inserter | 18 | 4 | 328 | 453 | 58 |
| 16 | Virtual Delay inserter | 8 |  | 190 | 261 | 38 |
|  | Delay Synchronizator | 8 |  | 127 | 108 | 20 |
|  | Inserter | <1% | 4 | 2 | 2 |  |
|  | AXIS Register Slice | 2 |  | 9 | 82 |  |
|  | Time Killer | 7 |  | 35 | 159 |  |
|  | Node Inserter | 18 | 6 | 409 | 632 | 72 |
| 24 | Virtual Delay inserter | 8 |  | 259 | 392 | 48 |
|  | Delay Synchronizator | 8 |  | 140 | 174 | 24 |
|  | Inserter | 1 | 6 | 2 | 2 |  |
|  | AXIS Register Slice | 1 |  | 8 | 64 |  |
|  | Time Killer | 7 |  | 72 | 223 |  |

**Table 3.** *Cont.*

| $N_{bit}$ | Module/Submodules | Power [mW] | CARRY | LUT | FF | LUTRAM |
|---|---|---|---|---|---|---|
| | Node Inserter | 23 | 8 | 429 | 785 | 84 |
| 32 | Virtual Delay inserter | 10 | | 259 | 497 | 56 |
| | Delay Synchronizator | 10 | | 159 | 204 | 28 |
| | Inserter | 1 | 8 | 2 | 2 | |
| | AXIS Register Slice | 2 | | 9 | 82 | |
| | Time Killer | 7 | | 94 | 415 | |
| | Node Inserter | 27 | 10 | 549 | 937 | 108 |
| 40 | Virtual Delay inserter | 12 | | 314 | 601 | 72 |
| | Delay Synchronizator | 12 | | 224 | 236 | 36 |
| | Inserter | <1% | 10 | 2 | 2 | |
| | AXIS Register Slice | 3 | | 9 | 98 | |
| | Time Killer | 10 | | 101 | 407 | |
| | Node Inserter | 30 | 12 | 633 | 1090 | 120 |
| 48 | Virtual Delay inserter | 13 | | 368 | 706 | 80 |
| | Delay Synchronizator | 13 | | 254 | 258 | 40 |
| | Inserter | <1% | 12 | 2 | 2 | |
| | AXIS Register Slice | 4 | | 9 | 124 | |
| | Time Killer | 16 | | 126 | 479 | |
| | Node Inserter | 33 | 14 | 716 | 1242 | 132 |
| 56 | Virtual Delay inserter | 14 | | 423 | 810 | 88 |
| | Delay Synchronizator | 14 | | 282 | 300 | 44 |
| | Inserter | 1 | 14 | 2 | 2 | |
| | AXIS Register Slice | 4 | | 9 | 130 | |
| | Time Killer | 18 | | 140 | 415 | |
| | Node Inserter | 37 | 16 | 802 | 1395 | 144 |
| 64 | | 16 | | 477 | 915 | 96 |
| | Delay Synchronizator | 16 | | 314 | 329 | 48 |
| | Inserter | <1% | 16 | 2 | 2 | |
| | AXIS Register Slice | 5 | | 9 | 149 | |
| | Time Killer | 20 | | 161 | 535 | |

## 6. Measures and Characterizations

A 3- and 16-channel TDC IP-Cores (with 3 and 16 parallel outputs each), provided by TEDIEL S.r.l. [24], was utilized to test the entire system and undertake the validation of what has been proposed. In Table 4, all the performance metrics of the two TDCs are reported, identical in all respects except for the number of channels. Obviously, the ODR is expressed as the output rate of each individual channel, which will be modified by subsequently inserting the Node Inserter and the structure of the BB.

Tests on the reference TDC architectures are performed on different FPGAs. The host FPGAs are both Xilinx 28 nm 7-Series: an Artix-7 100T for the 3-channel TDC (Figure 14) and a Kintex-7 325T for the 16-channel solution (Figure 15).

**Table 4.** TDC performance.

| Feature | Value |
|---|---|
| Number of Channels | 3 and 16 |
| $N_{bit}$ | 32 |
| LSB | 36.6 fs |
| FSR | 157.3 μs |

**Table 4.** *Cont.*

| Feature | Value |
|---|---|
| $f_{ovfl}$ | 6.36 kHz |
| Dead Time | 5 ns |
| Maximum Channel Rate | 120 MHz |
| ODR/Ch | 120 Msps |
| Precision | <12 pr r.m.s. |
| DNL | <800 fs |
| INL | <16 ps |
| LUT/Channel | 3869 |
| FF/Channel | 5255 |
| LUTRAM/Channel | 75 |
| CARRY/Channel | 390 |
| BRAM/Channel | 2 |
| Power/Channel | 284 mW |

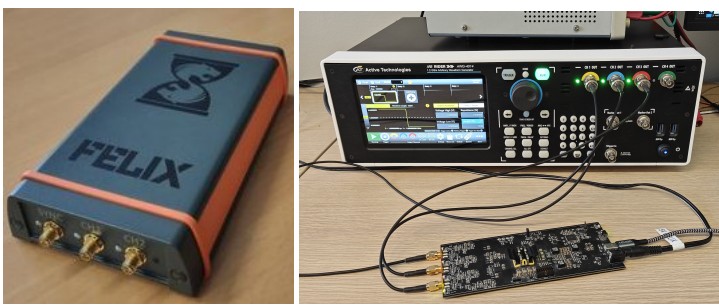

**Figure 14.** Picture of the FELIX board (**left**) hosting the Artix-7 100T for the 3-channel TDC IP-Core and the setup (**right**).

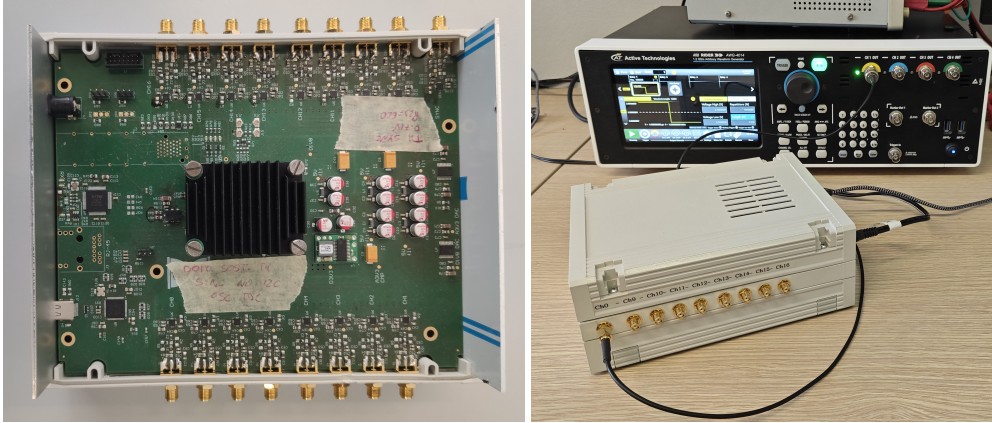

**Figure 15.** Picture of the Panther (**left**) board hosting the Kintex-7 325T for the 16-channel TDC IP-Core and the setup (**right**).

The performed tests involve injecting a pseudo-random signal into each channel of the TDC using the ACTRIVE Arbitrary Function Generator (AWG). Since the AWG has 4 channels in the 16-channel solution, it was decided to divide each channel of the AWG so that it controls 4 channels of the TDC. Pseudo-random signals were generated to ensure a distance between successive events greater than the set dead time. The experiment was conducted by uniformly increasing the rates of all TDC channels, monitoring and subsequently analyzing the output of the BB to verify its correct operation. The status of the various nodes and the Order Checker was also monitored to verify the occurrence of discard phases and the absence of unordered timestamps. To validate the correct operation

of the Time Killer and Virtual Delay modules, the experiment was automated with a script and repeated for numerous values of dead time (1024) and virtual delay (512).

Another test performed, once again with the help of the AWG and a script for its automation, both on the 3-channel and 16-channel versions, involved keeping the rate of all channels except one at zero and then increasing it uniformly, while monitoring and subsequently analyzing the output of the BB and the Order Checker. The experiment was repeated for each channel, always with 1024 values of dead time and 512 of virtual delay.

Outcomes of the experiments are reported in Table 5 for the 3-channel TDC and in Table 6 for the 16-channel solution.

For each of the tests described before, the absence of unsorted timestamps at the BB output was firstly checked. Even though the value of the Order Checker counter is always 0, this module has been kept in for safety reasons; mainly because, although it is simple to model during simulation, the clock uncertainty caused by the CDC is high in rare cases in real situations. Another milestone concerns the ODR, which can achieve up to 149.9 Msps and 199.9 Msps for Artix-7 and Kintex-7, respectively. Additionally, the presence or absence of at list one Discard Phase were monitored; it was observed when the sum of the total rates across the 16 channels reached 97% of the ODR in both solutions.

**Table 5.** BB performance in presence of different versions of BB proposed in this work for the 3-channel TDC.

| Feature | Section 3 | Section 4 | Section 5 |
|---|---|---|---|
| Number of Channels | 3 | 3 | 3 |
| $N_{bit}$ | 32 | 32 | 32 |
| LSB | 36.6 fs | 36.6 fs | 36.6 fs |
| FSR | 157.3 µs | 157.3 µs | 157.3 µs |
| $f_{CLK,BB}$ | 130 MHz | 150 MHz | 150 MHz |
| $f_{ovfl}$ | 6.36 kHz | 6.36 kHz | 6.36 kHz |
| ODR | 129.9 Msps | 149.9 Msps | 149.9 Msps |
| Rate w/o Discard | 126.0 Msps | 146.0 Msps | 145.0 Msps |
| Dead Time | 5 ns | 5 ns | 5 ns ÷ 1 ms |
| Virtual Delay | N.A. | N.A. | 0 ÷ 78.6 µs |
| Unsorted Timestamp | 1% | N.A. | N.A. |
| BB Total Occupancy LUT | 0.96% | 0.96% | 2.43% |
| BB Total Occupancy FF | 0.49% | 0.69% | 1.79% |
| BB Total Occupancy LUTRAM | 0.91% | 0.91% | 2.74% |
| BB Total CARRY | 0.30% | 0.15% | 0.15% |
| TDC Total Occupancy LUT | 18.5% | 18.5% | 18.5% |
| TDC Total Occupancy FF | 12.6% | 12.6% | 12.6% |
| TDC Total Occupancy LUTRAM | 0.55% | 0.55% | 0.55% |
| TDC Total CARRY | 7.43% | 7.43% | 7.43% |
| TDC Total BRAM | 4.39% | 4.39% | 4.39% |
| BB Total Power | 144 mW | 208 mW | 480 mW |
| TDC Total Power | 4544 mW | 4544 mW | 4544 mW |

In both solutions, the ODR, being $f_{ovfl}$ negligible as per (6), is very close to $f_{CLK,BB}$, which is, respectively, 150 MHz for the Artix-7 solution and 200 MHz for the Kintex-7, representing 24% and 25% of the maximum clock frequency that the two devices can handle (625 MHz for the Artix-7 and 800 MHz for the Kintex-7). This is an excellent result considering that, typically, the maximum clock frequency of a system in an FPGA is between 10% and 15% of the maximum frequency. Moreover, from the perspective of area utilization and power consumption, we observe that the presence of the BB is negligible (at least a factor 10) compared to that of the TDC.

Furthermore, the dependence of the maximum clock frequency of the BB (expressed as a ratio to the maximum frequency allowed by the FPGA, $f_{FPGA}^{MAX}$) on the number of nodes/channels ($N_{CH}$) and the number of bits ($N_{bit}$) on different devices of the Artix-7 family (i.e., 35T and 100T with 32,280 and 101,440 logic cells, respectively, and a maxi-

mum clock frequency of 625 MHz) and Kintex-7 (i.e., 325T and 480T with 326,080 and 477,760 logic cells, respectively, and a maximum clock frequency of 625 MHz) was analyzed by compiling different versions of the BB based on $N_{bit}$ and $N_{CH}$.

**Table 6.** BB performance in presence of different vertions of BB proposed in this work for the 16-channel TDC.

| Feature | Section 3 | Section 4 | Section 5 |
|---|---|---|---|
| Number of Channels | 16 | 16 | 16 |
| $N_{bit}$ | 32 | 32 | 32 |
| LSB | 36.6 fs | 36.6 fs | 36.6 fs |
| FSR | 157.3 μs | 157.3 μs | 157.3 μs |
| $f_{CLK,BB}$ | 130 MHz | 200 MHz | 200 MHz |
| $f_{ovfl}$ | 6.36 kHz | 6.36 kHz | 6.36 kHz |
| ODR | 129.9 Msps | 199.9 Msps | 199.9 Msps |
| Rate w/o Discard | 126.0 Msps | 193.0 Msps | 192.0 Msps |
| Dead Time | 5 ns | 5 ns | 5 ns ÷ 1 ms |
| Virtual Delay | N.A. | N.A. | 0 ÷ 78.6 μs |
| Unsorted Timestamp | 1% | N.A. | N.A. |
| BB Total Occupancy LUT | 1.58% | 1.58% | 3.98% |
| BB Total Occupancy FF | 0.80% | 1.08% | 2.94% |
| BB Total Occupancy LUTRAM | 1.50% | 1.50% | 4.50% |
| BB Total CARRY | 0.50% | 0.25% | 0.25% |
| TDC Total Occupancy LUT | 30.4% | 30.4% | 30.4% |
| TDC Total Occupancy FF | 20.6% | 20.6% | 20.6% |
| TDC Total Occupancy LUTRAM | 0.9% | 0.9% | 0.9% |
| TDC Total CARRY | 12.2% | 12.2% | 12.2% |
| TDC Total BRAM | 7.2% | 7.2% | 7.2% |
| BB Total Power | 144 mW | 208 mW | 480 mW |
| TDC Total Power | 4544 mW | 4544 mW | 4544 mW |

The results, shown in Figure 16, highlight a dependence of the ratio $\alpha$ defined as $f_{CLK,BB}/f_{FPGA}^{MAX}$ (where $f_{FPGA}^{MAX}$ is 625 MHz for Artix-7 and 800 MHz for Kintex-7) on the product $N$ defined as $N_{CH} \times N_{bit}$; we can observe a drop in $\alpha$ when the ratio between Logic Cells (LCs) and $N$ is below a value roughly between 300 and 500. This trend is due to routing difficulties caused by the reduction in available resources, indicated by the number of Logic Cells (LC) provided by the device, and the linearity with which the internal modules of the BB scale in terms of area occupation.

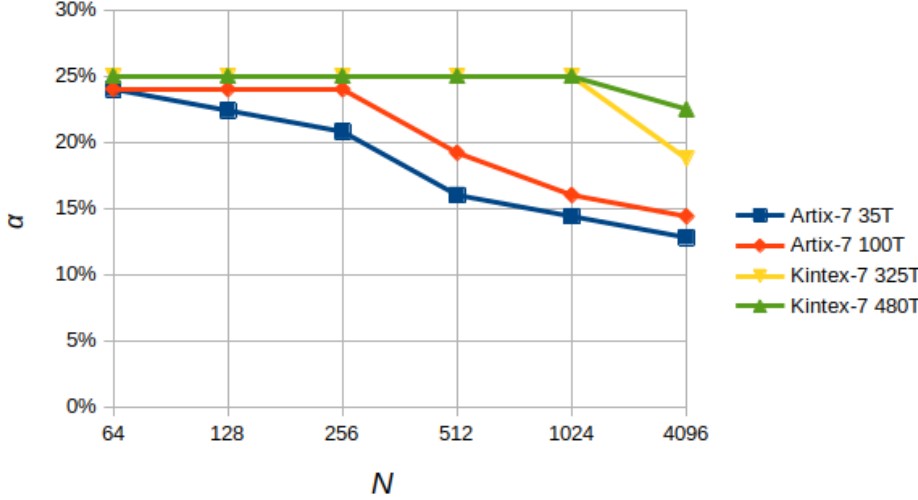

**Figure 16.** Picture of the Panther board hosting the Kintex-7 325T for the 16-channel TDC IP-Core.

## 7. Conclusions

This work focuses on a new timestamp management system called BB. The objective is to implement a parallel-to-stream conversion to alleviate the routing of timestamps from high-performance TDCs (i.e., high resolution and FSR, resulting in a high number of bits at a high rate) to the processing module in multichannel applications. The key characteristic of this method is the serialization of several TDC channels in a modular approach, producing timestamps in chronological order while flagging overflow.

In this paper, two issues have been addressed, and two new functionalities have been introduced. The first issue pertains to the occurrence of unsorted timestamps due to the CDC between the clock of the TDC and the BB. Subsequently, by enhancing the FPGA's critical path operations, a second issue related to the Belt-Bus's restricted output rate was resolved.

Additionally, two new features have been incorporated. The first is a Virtual Delay, utilized to compensate for offsets resulting from varying wire lengths between TDC channels and mismatched FPGA routing circuits. The second is Virtual Dead Time, employed to eliminate unforeseen events caused by residual noise at the TDC input.

The BB has been tested on a Xilinx 28 nm 7-Series Kintex-7 325T FPGA, yielding an overall data rate of 199.9 Msps with very limited resource usage (i.e., less than a total of 4.5%) and a power consumption of only 480 mW, considering a 16-channel implementation.

**Author Contributions:** Methodology, F.G.; Software, G.B.; Validation, E.R. and A.C.; Writing—original draft, N.L.; Writing—review & editing, A.G. All authors have read and agreed to the published version of the manuscript.

**Funding:** This research received no external funding.

**Data Availability Statement:** The data presented in this study are available in this article.

**Acknowledgments:** A special thanks goes to TEDIEL S.r.l., a spin-off of Politecnico di Milano, for providing the TDC IP-Core.

**Conflicts of Interest:** The authors declare no conflicts of interest.

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
