# Peer review of "New High-Rate Timestamp Management with Real-Time Configurable Virtual Delay and Dead Time for FPGA-Based Time-to-Digital Converters"

_electronics, doi:10.3390/electronics13061124_

Round 1

Reviewer 1 Report

Comments and Suggestions for Authors

This paper presents the “Built-Bus”, with the intended purpose of serializing timestamp data coming from multiple TDC channels while trying to maintain the chronological order of the timestamps. The scope of the work is quite narrow as it only presents a simple Parallel-in Serial-out interface for a muti-channel TDC core without the design of an actual TDC IP core.

Below are some major concerns that I feel should be addressed by the authors:

1)       After the Introduction section, there should be a Background section formally describing TDC and the inner workings of a basic TDC core. It is important for such background information to be presented early in the paper in order for the reader to understand the purpose of the presented work. There are some parts briefly describing TDC scattered in different sections of the paper, these parts should be removed and gathered in a background section that contains more details.

2)       There should be another section dedicated to the related work. In particular, the authors should discuss existing FPGA implementations of TDC systems, and most importantly show the need for the proposed Built-Bus. The main objective of the Built-Bus is to circumvent the “routing challenges” when connecting multi-channel TDC cores to processing modules. These challenges should be demonstrated to the reader. 

3)       Section 5 (experimental work) is very brief, lacking substantial important information for the reader (more details about the used TDC IP, experimental setup, how measurements are performed, how results are recorded and analyzed, etc.).  There is an image of an FPGA board without any connections or any setup!  

Below are some of the points that require further clarification:

1)       You mentioned that the bus is based on the AXI4-Stream with only TVALID and TDATA signals. Is there no TREADY signal? To be compliant with AXI4-Stream protocol the TREADY signal is required to allow a consumer to backpressure when not ready to receive data.

2)       In Section 2.2, it is not clear what is the difference between Retain and Hold-on phases.

3)       In Discard phase the timestamp is said to be discarded because the chain is full. What impact would this have on the system? How often timestamps get discarded? Also, there is no mention of any discarded timestamped in Section 5.

4)       In Figure 5, there is no FIFO at the Top connection, is this correct? If the inserter is passing timestamps from the Left connection, shouldn’t the timestamps from the Top connection be buffered?

5)       You should show the effect of increasing the number of nodes (channels) in the bus on the maximum operating frequency. Only 16 channels are used in Section 5. Would the maximum operating frequency be affected when the number of channels is increased? Also show max. frequency for larger sizes of Nbit.

6)       Is 200MHz considered good for Kintex-7 FPGA family?

7)       Power dissipation numbers are reported in milli Watts. Are these results good or bad for such applications? In a complete system, would the bus power dissipation be significant considering the overall power dissipation? Do the reported numbers consider both dynamic and static power?  

Minor issues:

1)       Revise and fix typos. Examples:

·       Section 2.2: “Lest” è Left ,

·       Section 4.2: “Node Inserte” è Node Inserter

·       Section 3.2: “occupe” è occupy

·       Axis Register Slice? Do you intend AXI4 or Axis?

2)       Consider putting more thought into choosing the names for the sub-modules in the design (Time Killer, Virtual Delay, etc.)

Comments on the Quality of English Language

The English is fine. Minor typos detected.

Author Response

Dear reviewer,

thanks for the revision; in attached the answers to your majors, concerns and minors.

Reviewer 2 Report

Comments and Suggestions for Authors

The article presents data and timestamp management system implemented in FPGA for acquisition of data coming from TDC converters. I have following comments:

1. You refer (line 20-24,  49-52), among others, to time-resolved experiments such as pump-probe, that take advantage of single photon counting detectors, and detectors with TDC with large number of channels. Some of them already include some kind of event-driven algorithm (e.g. Timepix detector) or compression algorithm, e.g. zero-supression, implemented inside the detector/IC. Please, add some references to the text and briefly discuss when your solution is better (line 55).

2. Additionally, there are existing solutions that implements event-driven or compression algorithms in FPGA, while also preserving timestamping. Please, add some references to the text.

3. Line 158 - "In dettail" double "t"

4. Line 403 - there is redundant space before the dot

5. Sometimes you refer to table as Table (e.g. line 170) and sometimes as Tab (line 300), same with Figures. I'd suggest to stick to one convention.

Although I find the paper more engineering than scientific, I think the presented solution description can be interesting to the engineers that are looking for the solution for managing TDC output timestamps in FPGA with precise, configurable delay correction.

Author Response

Dear reviewer 2,

thanks for the revision; in attached the answers to your issues.

Round 2

Reviewer 1 Report

Comments and Suggestions for Authors

The structure of the revised version of this paper is much better compared to the original version. All my suggestions have been addressed.

Please proofread the paper carefully before final submission. I have detected other typos in the second round of review:

1)       Table 5 caption: different version of BB should be different version(s) of BB

2)       Table 6 caption: different vertion of BB should be different versions of BB

3)       I can’t find references 17 to 20 and 22 to 24.

Comments on the Quality of English Language

Minor editing required.

Author Response

Thanks for the revision.

  1. Table 5 caption: different version of BB should be different version(s) of BB

    Done, thanks.

  2. Table 6 caption: different vertion of BB should be different versions of BB

    Done, thanks.

  3. I can’t find references 17 to 20 and 22 to 24.

    Done, thanks.